# SUNMASK: Mask Enhanced Control in Step Unrolled Denoising Autoencoders

## Abstract

This paper introduces SUNMASK, an approach for generative sequence modeling based on masked unrolled denoising autoencoders. By explicitly incorporating a conditional masking variable, as well as using this mask information to modulate losses during training based on expected exemplar difficulty, SUNMASK models discrete sequences without direct ordering assumptions. The addition of masking terms allows for fine-grained control during generation, starting from random tokens and a mask over subset variables, then predicting tokens which are again combined with a subset mask for subsequent repetitions. This iterative process gradually improves token sequences toward a structured output, while guided by proposal masks. The broad framework for unrolled denoising autoencoders is largely independent of model type, and we utilize both transformer and convolution based architectures in this work. We demonstrate the efficacy of this approach both qualitatively and quantitatively, applying SUNMASK to generative modeling of symbolic polyphonic music, and language modeling for English text.

## 1 Introduction

Generative modeling approaches can stratified into different modeling approaches based on factorization to form two broad categories, autoregressive modeling (AR) and non-autoregressive modeling (NAR). We introduce SUNMASK, a NAR generative model for structured sequences.

### 1.1 Autoregressive Models

AR modeling with deep neural networks has been a dominant approach to generative modeling and feature learning [38, 70, 73, 39, 76, 74] which has many crucial advantages in both training and inference. One key concern is the necessity of defining a "dependency chain" in the form of a (typically) directed acyclic graph (DAG). Sampling during inference can be accomplished in a straightforward manner using ancestral sampling - sampling from the first variable or variables in the DAG, using those to conditionally estimate a probability distribution for subsequent variables.

Many applications have straightforward orderings in which to define this chain of variables, based on domain knowledge. For example following the flow of time for timeseries modeling is often a logical choice, allowing models to make predictions into the future from the past. However in many other domains, for example images, language, or music, the process of defining a dependency chain over input variables (e.g. pixels, characters, words, or notes) is far from straightforward, as for any arbitrary ordering there can frequently be examples where this ordering *creates* long-term dependencies, or otherwise makes satisfaction of dependencies during training and evaluation more difficult than another alternative ordering.

This divide becomes further compounded in many creative applications to these domains, as creators typically iterate repeatedly: forming a concept, applying an initial sequence of steps to create the

framing of the concept, and seeing where the creative flow may lead to alterations in the original concept, thus altering future steps. Though the resulting output may be perceived in a time-ordered fashion (for example, reading a book or listening to a song), the initial creation was performed globally and holistically. This global view is often critical to creating elements such as foreshadowing and tension which make the resulting output interesting or enjoyable. This iterative process is directly at odds with a strict AR factorization, and requires well trained AR models to cope with a high degree of uncertainty and multi-modality for long range dependencies, which can lead to logical mistakes or other errors.

## 1.2 Non-Autoregressive Models

An alternative methodology for generative modeling is non-autoregression (NAR), broadly covering a large number of different modeling approaches which attempt to remove assumptions about variable ordering, instead either hand-defining per-exemplar orderings, or modeling variables jointly without resorting to chain rule factorization. One way to define an ordering over variables is via masking of inputs or intermediate network representations [22, 71, 72, 77, 73, 57], and indeed modern AR approaches such as transformers [75] use an autoregressive mask internally to define the chain of variables order. These masks can either be constant over all training (as in standard AR transformers and PixelCNN [73]) or dynamic per example (as in MADE [22]). When masks are dynamic per example, we begin to see the relationship between enforcing AR via masking and NAR methods, as although some ordering is assumed this ordering is no longer constant, and it becomes possible to use the same trained model to evaluate the probability of a particular output variable under *multiple* possible orderings.

Closely linked to masking methods are so called *diffusion models*, which relax the variable ordering problem through noise prediction [67, 69, 30]. Rather than predicting a new variable or variables given previous ones in an arbitrarily chosen DAG, diffusion models focus on predicting a less noisy version of many variables jointly, given a set of noisy input variables. Iteratively applying this learned denoising improvement operator should eventually result in predicting a fully clean output estimate, given either a noisy version of the target domain, or even starting from pure noise. Given this framing it is clear that diffusion models are closely linked to denoising methods in general, specifically denoising autoencoders, as well as modern density modeling approaches such as generative adversarial networks (GAN [23]), variational autoencoders (VAE [41]), flow-based models (NICE [15], RealNVP [16], Normalizing Flows [61], IAF [42], MAF [57]), iterative canvas sampling (DRAW [24]), and noise contrastive estimation (NCE [27]). Particular applications of this denoising philosophy such as BERT [14], WaveGrad [9], and GLIDE [56], have resulted in large quality improvements for feature learning and data generation for text, images, and audio [46, 66, 60, 29].

## 1.3 Trade-offs Between AR and NAR Approaches

The choice between AR and NAR methods is not clear-cut. For many domains, high-quality models exist using both approaches but we can define some crucial parameters. Some NAR methods such as GAN or VAE are capable of generating output in only one inference step, however they are typically hard to train on certain data modalities (e.g. text data) comparing to AR counterparts. Other NAR methods such as diffusion models typically allow for choosing a diffusion length during inference, which is independent of that used at training. Choosing a low diffusion length can frequently lead to poor sample quality, and tuning this setting (among many others) is critical to high quality generation. However if the tuned diffusion length for a given sequence (of length $T$) is shorter than the length of those sequences, the NAR method has a computational advantage over the equivalent AR model (which would require $T$ steps for a $T$ length sequence). In addition, the ability to tune this diffusion length can be useful in interactive applications, or when a variety of output is desirable. This setting can also be a curse, as even well-trained models perform poorly with improper diffusion settings. Several branches of current research are focused on improving guarantees [35, 68, 40] and convergence speed for diffusion models [43, 44, 36, 79].

## 1.4 SUNMASK, a non-autoregressive sequence model

We introduce SUNMASK, a NAR sequence model which uses masks over noised, discrete data to learn a self-improvement operator to transition from categorical noise to the data distribution in

iterated steps. Given a target data representation, we train a model which can map from a noisy version of input data to a corrected form of the input. In this work, we use multinomial noise - namely entries are corrupted to 1 of $P$ possible values (for a given set size $P$), with the number of noised entries defining the relative noise level for the training example. This is similar to many diffusion approaches at a high level, and particularly shown to be an effective tool in SUNDAE [65] and Coconet [33]. In addition to the use of multinomial noise, we also form a mask representing *where* the data was noised, feeding this mask alongside the input data to form a conditional probability distribution.

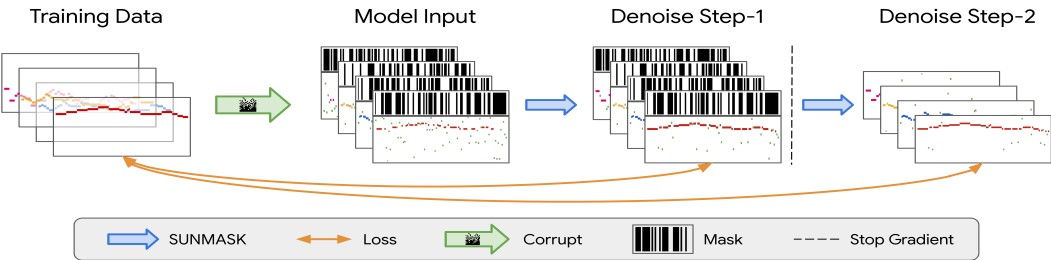

Figure 1: Step-unrolled denoising training for SUNMASK on polyphonic music, unrolled step length 2. Training data (left) consists of four voices corrupted by sampling a random mask per voice and replacing the masked data (red) with random pitches (green). SUNMASK takes both mask and corrupted training data as input, predicting denoised original data as output. In the second step, the model takes a sampled version of the model step predictions and the same mask as input, outputting another prediction of the original data.

## 2 Method

The relationship between discrete diffusion and denoising autoencoders has been explored in previous work [31, 65, 3, 32]. We build upon this foundation, combined with many insights from prior orderless modeling approaches, crucially Orderless NADE [72], Coconet [33] (which is a more modern variant of Orderless NADE), and SUNDAE [65].

SUNMASK is built around a process $x_t \sim f_\theta(\cdot|x_{t-1}; m)$ on a space $X = \{1, \ldots, v\}^N$ of arrays of categorical variables. This parametric transition function $f_\theta$ takes an additional argument $m \in {0, 1}^N$. During training, $m$ indicates variables that were not corrupted, and as a consequence we can use it during inference to tell $f_\theta$ which variables to trust.

Given a sequence of masks $m_0, \ldots, m_{T-1}$, the generating distribution of our model derives from a prior $p_0$ (typically uniform noise) and repeated application of $f_\theta$:

$$p_T(x_T; m_0, \ldots, m_{T-1}) = \left( \sum_{x_1, \ldots, x_{T-1} \in X} \prod_{t=1}^{T} f_\theta(x_t|x_{t-1}; m_{t-1}) \right) p_0(x_0) \qquad (1)$$

In practice, $p_0$ is typically elementwise iid uniform noise, and the masks $m_0, \ldots, m_{T-1}$ are drawn according to a schedule and may be held constant for several steps.

To train $f_\theta$, we take a training example $x \sim p_{\text{data}}$ and draw a mask $m$. We apply the corruption procedure $x_0 \sim q(\cdot|x; m)$ to obtain $x_0$ which equals $x$ where the mask $m$ is true and uniform random values elsewhere. Then we iterate $x_t \sim f_\theta(\cdot|x_{t-1}; m)$ with the aim of reconstructing $x$.

As in SUNDAE, the transition $f_\theta$ models the variables as conditionally independent of one another. However SUNDAE has no direct concept of masking. SUNMASK thus combines past insights from the masked NAR models Orderless NADE and Coconet with existing concepts from SUNDAE, along with new model classes and inference schemes to form a powerful generative model. Similar to SUNDAE, our objective is to minimize $\frac{1}{2}(L^{(1)} + L^{(2)})$ where

$$L^{(t)}(\theta) = -\mathbb{E}_{m_0, \cdots, m_{t-1}} \mathbb{E}_{\substack{x \sim p_{\text{data}} \\ x_0 \sim q(\cdot|x, m_0) \\ x_1 \sim f_\theta(\cdot|x_0; m_0) \\ x_2 \sim f_\theta(\cdot|x_1; m_1) \\ \cdots \\ x_{t-1} \sim f_\theta(\cdot|x_{t-2}; m_{t-2})}} \left[ \frac{\sum_i (1 - m_{t-1}^{(i)}) \log f_\theta^{(i)}(x^{(i)}|x_{t-1}; m_{t-1})}{\sum_i 1 - m_{t-1}^{(i)}} \right]$$

$$(2)$$

is the reconstruction loss for the elements of $x$ that were corrupted. As in Orderless NADE [72] and Coconet [33], we weigh each term according to the size of the mask, to ensure that the overall weight on each conditional $f_\theta^{(i)}$ is uniform across $i$. Unlike previous methods, we target *only masked variables* in the loss. In practice we choose $m_0 = \cdots = m_{t-1}$ during training and $t = 2$. Since we only go to $t = 2$, keeping the mask constant is a close enough approximation to the masking schedule used in inference. The choice of $t = 2$ is driven by the ablation study in SUNDAE, where $t = 2$ was found to account for nearly all performance gains in translation experiments, with higher unrollings showing no additional benefit. In addition higher values of $t$ unrolling generally increase memory usage, making the training of high order unrollings complicated.

Coupled with multi-step unrolling, the SUNMASK training scheme encourages learning complex relationships between the mask and the data, allowing the potential for multi-level trust over the input data: variables with a mask value of 1 which appear correct (given context), variables of mask value 1 which appear incorrect, variables of mask value 0 which appear correct, and variables of mask value 0 which appear incorrect. Denoising only methods (such as SUNDAE [65]) would need to form an internal, non-controllable mask in order to disentangle these states, and 0 mask models (such as Coconet [33]) have controllable input masks but combine both masked states.

SUNMASK allows for direct control at inference using both proposal masks and noising of variables, combining elements of both SUNDAE and Coconet. We show a high level example of the unrolled training scheme, mask proposals, and input data processing in Figure 1.

## 2.1 SUNMASK, SUNDAE, and Coconet Comparison

The overall unrolled mask and iterative inference setting is largely independent of architecture choice, and as long as the internal architecture does not make any ordering assumption over the input data we can incorporate it into SUNMASK. We use two primary archetypes for the internal model in this paper: Attentional U-Net and Relative Transformer. Detailed description of the respective architectures can be seen in the Appendix.

SUNMASK uses an unrolled training scheme, similar to that shown in SUNDAE, as well as a mask which is input to the model and defines manipulated variables as in Coconet. The loss is masked based on this manipulation mask, unlike Coconet or SUNDAE. The SUNMASK loss is further weighted by the total amount of masked variables. Comparisons of various high level modeling features between SUNMASK, Coconet, and SUNDAE are shown in Table 1.

Table 1: Comparison of model features for
SUNMASK, Coconet, and SUNDAE

| Model | SUNMASK | Coconet | SUNDAE |
|---|---|---|---|
| Mask input to model | ✓ | ✓ | ✗ |
| Masked loss | ✓ | ✗ | ✗ |
| Re-weighted loss | ✓ | ✓ | ✗ |
| Unrolled loss | ✓ | ✗ | ✓ |
| Inference mask schedule | ✓ | ✓ | ✗ |
| Sampling rejection step | ✓ | ✗ | ✓ |
| Mask control preserves data | ✓ | ✗ | ✗ |

## 2.2 Model Training

During training, the internal architecture is combined with a *step unrolled* training procedure, as highlighted by SUNDAE [65]. Rather than directly randomizing positions, we re-write this as a masking scheme, first sampling a mask (with 0 randomize, 1 keep, which we denote as 0-active format) then performing randomization to one of $P$ possibilities, for the masked subset of $K$ variables. This random masking procedure is equivalent to the approach from SUNDAE, but using a mask allows us to further combine the mask information with the input data, in order to form a conditional probability estimate. In addition, this 0-active masking scheme makes direct comparison to masking schemes with absorbing states (such as OrderlessNADE [72], Coconet [33], VQ-Diffusion [25] and OA-ARDM [31]) simpler, as the mask can be directly multiplied with the data in a 0-active format.

Convolutional SUNMASK incorporates mask information with a one-hot data representation by concatenation along the channel axis. Transformer SUNMASK uses a slightly different setting -

assuming input transformer data $(T, B)$, with $T$ timesteps, $B$ batch elements, and vocabulary size $P$ is transformed to a $(T, B, L)$ dimensional embedding, we repeat the mask along the embedding axis $L$ times, downweighting the values in the mask by $\frac{1}{\sqrt{L}}$ for numerical reasons. Concatenating this reduced mask with the input embedding along the last axis is sufficient to form the desired conditional probability distribution. This stretched and reduced mask format provides more stable training than other schemes such as separately embedding the mask, then concatenating or summing with the input data embedding.

Each training batch is randomly sampled from the training dataset, and a corresponding noise value drawn from $rand(N)$ for $N$ examples in the minibatch. This per-example noise value is then used to derive a per-step mask over $T$ timesteps, by comparing noise $rand(N) < rand(N, T)$. During training, this means some examples have a high per-example noise value (e.g. .99), and thus many values masked and noised in the training, while other examples may have a low noise value (e.g. .01) drawn instead. Combined with a training loss which learns to denoise the input and focuses on imputing information about masked corrupted inputs, the overall model will learn a chain to go from more noisy data to less noisy step-wise, resulting in a learned improvement operator [32, 65].

This improvement operator can be applied to noisy data or pure noise, and iterate toward a predictive sample from the training distribution. See Multinomial Diffusion [32] and SUNDAE [65] for more detail on this proof, as well as fundamental work on denoising autoencoders [1]. In SUNMASK, we combine the mask used to noise the input with the input data itself, while modifying the loss to predict *only masked variables*. In addition, we downweight the loss by $\frac{1}{1+\sum 1-m_t}$ for each batch element, meaning that losses for heavily masked entries are downweighted compared to losses on examples with little masking, in a form of curriculum weighting based on expected estimation difficulty.

While a one step denoising scheme can be sufficient for learning the data manifold [3, 1], *unrolling* this denoising scheme into a multi-step process can have performance benefits. SUNMASK directly uses the unrolled loop scheme described in [65], using a step value of 2. For a detailed description of the step unrolled training scheme, see Appendix or the overview description from SUNDAE [65]. The masked and unrolled training can be seen as a container for any internal model which does not make ordering assumptions, and we utilize both convolutional U-Net (a variant of GLIDE [56] U-Net) and Relative Transformer [12, 34, 59] models for various experiments, shown in Section 4.

## 2.3 Inference Specific Settings

Well-trained SUNMASK models should be applicable to full content generation, as well as a variety of partially conditional generative tasks such as infilling and human-in-the-loop creation. Basic sampling involves creating a set of variables, with all variables randomly set to 1 of $P$ values in the domain (or partial randomization in the case of infilling) along with an accompanying mask, which is initially all 0 for full generation, or mixed 1s and 0s for partial generation tasks. Given this data and mask as input, the trained model then predicts a probability distribution over all possible $P$ values, for all variables. Despite the use of masked losses in training, we sample these prediction distributions for *all* variables. These predictions are then accepted or rejected from the original set, resulting in a new variable set. We then sample a new mask (based on a predefined schedule) and combine it with the initial mask, then iterate this overall process, updating at least some of the variables at each step.

During inference we use several key techniques to improve generative quality. We use typicality sampling [52] on the output probability distribution and a variable number of diffusion steps, on the order of 100 to 2000. Masks are randomly sampled using the schedule defined in [33] which linearly decreases the number of masked variables over time according to $\alpha_n = \max(\alpha_{\min}, \alpha_{\max} - \frac{n}{\eta N}(\alpha_{\max} - \alpha_{\min}))$ with $\alpha_{\min} = .001$, $\alpha_{\max} = .999$, and $\eta = 3/4$ as in [33], along with an optional triangular linear ramp-up and ramp-down schedule for the probability of accepting predictions from the model into the current variable set at each step, as shown in [65]. Active balance, by increasing the probability of updating variables which have been updated less often, is another inference time option. Variables can also be re-noised at any step, randomly resetting any variable with a 0 value for the mask at that diffusion step. Some problems (primarily symbolic music modeling) showed increased variability from active balance and re-noising, but the usefulness of these options is task dependent.

We caution that tuning hyperparameters for inference is critical to success, as improper settings can drastically lower the performance of SUNMASK, see Section 4 for variance over various inference

settings in different tasks. For human-in-the-loop applications, the existence of these controls can allow a number of fine-grained workflows to emerge, driven by expert users to create and curate interesting output [18, 11], demonstrated in Figure 3.

# 3 Related Work

We state here some key related approaches, as well as how our method differentiates from these previous settings. A number of recent publications on diffusion models and feature learning have incorporated masks as part of their overall training scheme [31, 7, 29], however these papers use masks for blanking, rather than as indicators over stochastic variables. Many infilling models [17, 37, 14, 50, 10], and masked image models [29] feature conditional modeling with a mask (blank) token, predicting the variables masked from the input for feature learning or generative modeling. XLNet [77] combines the infilling and autoregressive paradigms, learning arbitrary permuted orders over masked out variables, using blank-out masking and randomly generated autoregressive ordering similar to OrderlessNADE and Coconet. Conditional diffusion generators [53, 63, 64] and GAN generators [20] have the combination of mask indicators as well as preserving stochasticity of the masked variables. However these methods do not use an unrolled training scheme, and generally target image related tasks, with the notable exception of maskGAN. Many models use a concept of a working canvas, and do repeated inference steps for generation or correction of data [24, 4, 45, 21, 55], SUNMASK differs from these models due to architecture choices, training scheme, and loss weighting, as well as application domain [54, 58, 62, 25, 56, 60].

## 3.1 Convolutional SUNMASK

SUNMASK is most closely related to coconet [33] and SUNDAE [65]. Coconet (as an instance of OrderlessNADE), trains by sampling a random mask per training example, using this mask to set part of the input (in one hot format) to zero. The mask is further concatenated to the zeroed data along the channel axis, and this combined batch is passed through a deep convolutional network with small $3 \times 3$ kernels. Convolutional SUNMASK uses a downweighted loss over only variables masked in the input. However, SUNMASK additionally uses the unrolled training scheme, as well as a different inference procedure due to preserving the values of masked out variables during training and sampling.

Our best performing convolutional SUNMASK architecture takes hints from recent image transformer and vector quantized generators, exchanging the small kernels used in Coconet for extremely large kernels of shape $4 \times P$ over the time and feature dimensions, somewhat analogous to input patches, removing the model's translation invariance over the feature axis by setting kernel dimension equal to the total feature size. However this makes the number of parameters per convolutional layer extremely large. Convolutional SUNMASK adopts an attentional U-Net structure which reduces only across the time axis, modified from GLIDE [56], rather than the deep residual convolution network used by Coconet. Combined with the addition of step unrolled training, we are only able to train convolutional SUNMASK with a batch size of $1$ (expanded to effective batch size 2 due to step unrolling) on commodity GPU hardware with 16GB VRAM.

Due to the design choice of extremely large kernel sizes which depend on the size of the domain, we only use convolutional SUNMASK for polyphonic music experiments, see Section 4 for more details. Exact specification of the convolutional U-Net architecture can be seen in the Appendix.

## 3.2 Transformer SUNMASK

Transformer SUNMASK relates closely to the transformer used in SUNDAE. The architecture uses a relative multi-head attention [12, 34] and has no autoregressive masking. SUNMASK transformer also uses larger batch sizes, typically 20 or larger, though this is far smaller than the batch sizes seen in the experiments of SUNDAE. Sequence length and data iterator strategy were both a critical aspect for training transformer SUNMASK. We found short sequences (from 32 to 128) worked best, along with iteration strategies that were example based. In the language experiments, padding each example to some max length resulted in more stable training than the typical language modeling approach of treating the corpus as one long sequence and slicing into even sized chunks, or iterating sequentially. The stability gap between padded sequences and non-overlapping chunking became especially apparent at sequence lengths above 128 with transformer SUNMASK.

Table 2: Quantitative results from the Bach Mock grading function [19].
Lower values represent better chorales.

| Model | Note | Rhythm | Parallel Errors | Harmonic Quality | S Intervals | A Intervals | T Intervals | B Intervals | Repeated Sequence | Overall |
|---|---|---|---|---|---|---|---|---|---|---|
| Bach GT | 0.24 ±0.15 | 0.23±0.14 | 0.0±0.69 | 0.41±0.2 | 0.47±0.28 | 0.49±0.22 | 0.53±0.24 | 0.69±0.4 | 1.29±0.88 | 4.91±1.63 |
| BachMock | 0.37±0.22 | 0.26±0.14 | 2.16±3.22 | 0.54±0.31 | 0.53±0.35 | 0.71±0.34 | 0.73±0.38 | 0.89±0.68 | 1.86±2.81 | 8.94±4.64 |
| **SMc-T-BEST20-200** | 0.39±0.16 | 0.53±0.26 | 0.0±0.81 | 0.68±0.27 | 0.59±0.25 | 0.88±0.42 | 0.80±0.20 | 0.71±0.27 | 1.44±0.52 | **7.16±0.97** |
| AugGen |  |  |  |  |  |  |  |  |  | 8.02±2.92 |
| Coconet | **0.44±0.23** | 1.85±0.39 | 2.61±6.56 | 1.38±0.39 | **0.70±0.17** | **0.86±0.73** | **0.86±0.42** | **1.02±0.38** | 6.07±1.76 | 17.00±6.58 |
| SD | 0.59±1.82 | 0.93±0.84 | 6.42±4.11 | 0.98±0.67 | 1.17±5.09 | 2.65±4.08 | 1.57±5.68 | 2.57±3.28 | 2.45±2.39 | 23.25±21.45 |
| SD-T | 0.63±2.40 | 0.60±0.96 | 3.82±4.98 | 0.96±0.64 | 1.21±5.03 | 3.40±4.99 | 3.02±5.02 | 2.36±3.90 | 1.52±3.43 | 20.09±23.88 |
| SD-AT | 0.52±2.42 | 0.60±0.95 | 3.18±5.10 | 0.96±0.64 | 1.24±5.00 | 3.93±5.03 | 2.22±5.04 | 2.00±3.91 | 1.80±3.39 | 18.90±24.15 |
| SMc | 0.87±2.05 | 0.63±0.77 | 1.38±6.00 | 1.02±0.49 | 1.41±5.28 | 2.02±4.36 | 1.94±5.72 | 2.91±4.94 | 2.32±2.31 | 22.47±20.80 |
| SMc-A | 1.02±2.22 | **0.47±0.77** | 3.92±3.91 | **0.91±0.55** | 2.32±5.23 | 3.54±4.98 | 2.74±5.30 | 5.96±4.59 | 2.23±3.82 | 27.82±18.82 |
| **SMc-T** | 0.57±1.79 | 0.69±0.35 | 1.28±3.73 | 0.93±0.49 | 0.80±4.51 | 0.99±4.01 | 1.20±4.68 | 1.40±3.91 | 1.81±0.83 | **13.43±19.27** |
| SMc-AT | 0.66±1.90 | 0.55±0.29 | 2.76±3.63 | 0.94±0.47 | 0.91±4.11 | 1.10±4.00 | 1.26±4.26 | 1.45±4.56 | 2.05±0.96 | 16.50±17.96 |
| SMc-ATN | 2.24±2.36 | 0.58±0.49 | 6.82±4.81 | 1.56±0.54 | 6.46±4.14 | 8.51±4.43 | 7.21±4.28 | 7.60±3.11 | **1.47±1.02** | 43.85±18.41 |
| SMt | 3.00±1.85 | 0.74±0.90 | **0.00±1.95** | 1.64±0.70 | 8.94±4.66 | 6.49±4.99 | 8.47±5.58 | 7.72±4.41 | 3.10±2.97 | 42.87 |
| SMt-A | 3.00±1.85 | 0.74±0.90 | 0.00±1.95 | 1.64±0.70 | 8.94±4.66 | 6.49±4.99 | 8.47±5.58 | 7.72±4.41 | 3.10±2.97 | 42.87 |
| SMt-T | 3.74±2.16 | 0.58±0.56 | 0.00±2.56 | 1.73±0.73 | 8.75±4.62 | 6.22±3.99 | 8.05±4.73 | 7.95±4.49 | 2.35±1.79 | 46.21±17.30 |
| SMt-AT | 3.74±2.16 | 0.58±0.56 | 0.00±2.56 | 1.73±0.73 | 8.75±4.62 | 6.22±3.99 | 8.05±4.73 | 7.95±4.49 | 2.35±1.79 | 46.21±17.30 |

We list the hyperparameters for the transformer SUNMASK models in the Appendix. Transformer SUNMASK was trained on every dataset used in this paper, and we show performance in Section 4, as well as comparisons to convolutional SUNMASK on symbolic polyphonic music modeling. Both convolutional and transformer based SUNMASK use the Adam optimizer, with gradient clipping by value at 3. Inference hyperparameter types and general sampling strategies used are the same with both models, though specific hyperparameter values may change between datasets.

## 4 Experiments

### 4.1 Quantitative Results

We demonstrate the use of SUNMASK for polyphonic symbolic music modeling on the JSB dataset [2, 5]. The JSB dataset consists of 382 four-part chorales, originally written by Johann Sebastian Bach. These chorales are quantized at the 16th note interval, cut into non-overlapping chunks of length 128, skipping chunks which would cross the end of a piece. This processing results in a training dataset of 4956 examples, with each example being size $(4, 128)$. We train convolutional and transformer versions of both SUNMASK and SUNDAE for comparison, as well as the pretrained Coconet [33]. For polyphonic music, the quantized data was rasterized in soprano, alto, tenor, bass (SATB) order, as in Music Transformer [34] and BachBot [47], then chunked into non-overlapping training examples. Results are shown in Table 2. These results are evaluated on Bach ground truth data (Bach GT), BachMock Transformer (BachMock [19, 49]) (closely related to the decoder from VQ-CPC [28]), Coconet, SUNDAE (SD), and SUNMASK convolutional (SMc) and SUNMASK transformer (SMt). Model variants are indicated with Active Balance (A), Typical Sampling (T), and Noise (N).

The grading function used for evaluation, referred to as BachMock here, is designed specifically to correlate with expert analysis on Bach. In particular using this metric to choose correct examples in a paired comparison test, outperforms novice, intermediate, and expert listeners by varying margins [19]. This indicates that scoring well on the aggregate metric should correlate to high sample quality. The metric has many sub-parts, ranking various musical attributes crucial to codifying the style of J.S. Bach. AugGen [49] incorporated this metric into an iterative training and sampling scheme which improved final generative capability for a fixed model, showing the effectiveness of BachMock in practice for ranking machine generated samples. For every grading function in Bach Mock grading function, we show the median value and $\pm$ standard deviation as well as the overall grade, and lower values are better. We see the strongest results for convolutional SUNMASK with typicality sampling. Combined with final top-N ($N = 20$) selection out of a candidate set of 200 samples, the overall sample quality outperforms strong baselines. In addition, the massive performance gain from top-N selection indicates that the variance is likely driven by failures during sampling, rather than more fundamental modeling errors.

The EMNLP 2017 News dataset is a common benchmark for word-level language modeling [6], containing a large number of news article sentences [51]. Preprocessing steps collapse to sentences containing the most common 5700 words, resulting in a training set of $200k$ sentences with a test set of $10k$. The overall maximum sentence length is 51. Common processing for this dataset includes

padding all sentences up to this maximum length, different than the standard long sequence chunking commonly used in other language modeling tasks.

We show the results of several SUNMASK models for generating sentences similar to EMNLP2017News, comparing to benchmarks using the standard Negative BLEU/Self-BLEU evaluation [80, 6] over generated corpora of 1000 sentences in Figure 2. This set of scores, varied across temperature, is compared against baseline scores [48, 78, 8, 26, 13, 75], similar to the evaluation shown in SUNDAE [65]. These reference benchmarks used 10000 sentences to form performance estimates.

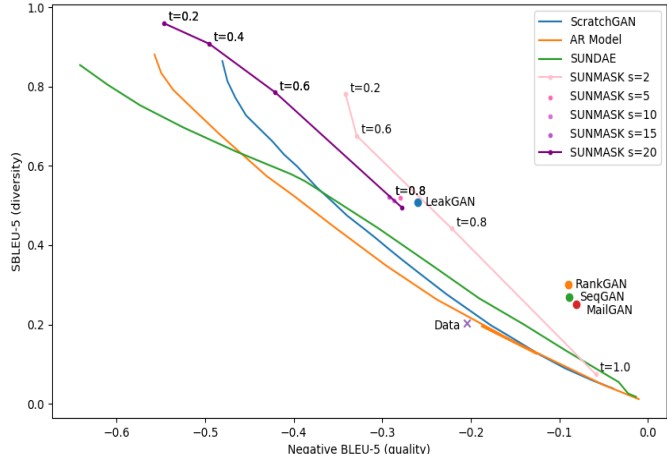

Figure 2: Negative BLEU/Self-BLEU scores on EMNLP2017 News. Left (x-axis) is better, lower (y-axis) is better. Quality/variation is controlled by changing the temperature (t), and varying diffusion schedule (s). For SUNMASK, *typical* sampling results [52] are shown.

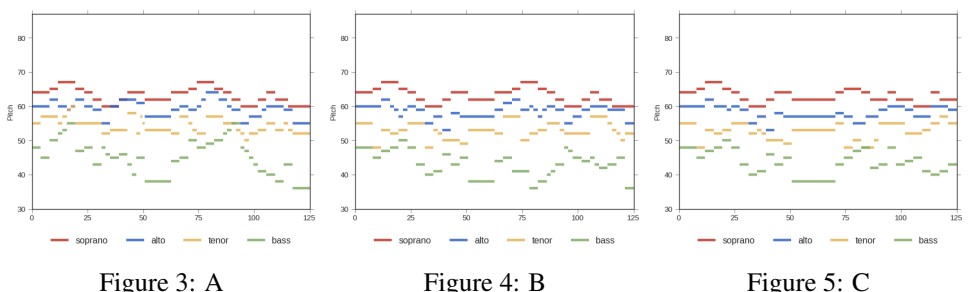

| Figure 3: A | Figure 4: B | Figure 5: C |

*SUNMASK* harmonization (bass, tenor, alto) of an existing melody (soprano)(A), with mask which highlights the left half (0 to 64) soprano voice (B), left half mask with right half replacement (C)

## 4.2 Qualitative Study of Masking For General Task Control

Given the flexibility of masking at inference, we perform a number of qualitative queries to inspect how the model adapts based on noise and mask value. Figures 3 4, and 5 demonstrate the use of SUNMASK for musical inpainting, holding the top voice (soprano) either fully or partially fixed to the well-known melody "Ode to Joy", by Ludwig van Beethoven. We see the trained model is more than capable of inpainting based on a pre-defined mask.

We test masks which hold the whole soprano fixed, masks which cover only parts of the soprano but do not allow randomization away from those notes, and masks which cover parts of the soprano but allow rewriting of the non-masked parts of the soprano, as well as rewriting all other voices. The use of masks to focus on subsets of variables while preserving underlying intermediate predictions is

unique to SUNMASK, as SUNDAE does not have an explicitly controllable input mask and Coconet does not have the ability to mask without also blanking the underlying variable.

This control is also demonstrated in Table 3, where masking is used to variably increase or decrease the weight on various pre-specified terms, held fixed throughout inference. The combination of these words, and their mask status can be seen to influence the overall tone of the selected text passages which showed the strongest effect in a particular inference batch. Though the generation quality is flawed, we clearly see a relationship between the masked word and the emergent surrounding context, for example highlighting **war** draws forth divorce, attack, and leave, while **play** instead references Premier Cup and excitement. We see similar results on a batch scale, and full demonstration of the text samples can be seen in the Appendix.

## 5  Conclusion

We introduce SUNMASK, a method for masked unrolled denoising modeling of structured data. SUNMASK separates the role of masking and correction by conditioning predictions on the mask, allowing for fine-grained control at inference. When applied to text as well as symbolic polyphonic music, SUNMASK is competitive with strong baselines, outperforming reference baselines on music modeling. Leveraging the separation of mask and noise allows for subtle control at inference, paving the way for a variety of domain specific applications and generative pipelines for human-in-the-loop creation.

Table 3: Qualitative samples using masks to emphasize the influence of particular words. Samples from a SUNMASK Transformer trained on the word level EMNLP2017News dataset.

| | |
|---|---|
| **Success** unmasked
*disaster* masked | I think I want to leave **success** at the end of the *disaster* , but because that ' s a nice to say it ' s not good to be the challenge and this is a very good thing \<eos\>
That was the job I was **success** to have to pay my *disaster* but hopefully I have been able to pull playing in the first couple of the season , I ' ve been happy to go through this team , he said \<eos\> |
| *Success* masked
**disaster** unmasked | Although more than 80 , 000 *success* have been displaced in the **disaster** since the last year , more than 700 , 000 lives have been injured in the country , and 70 of them were killed , according to the UN media \<eos\>
I haven ' t had a *success* at the league , the **disaster** and picked running with the door ago we have Champions , and I was a couple of pressure . . . and it was a lot of times \<eos\> |
| **Celebration** unmasked
*crime* masked | This is a fact on the **celebration** is such a really good *crime* , but it can be some of the most good people around the world and I think it must be the good way to do it \<eos\>
It ' s part of those **celebration** at the start of the *crime* , and it ' s a lot of good pride over the past few years , it ' s going to be more happy to play through this world \<eos\> |
| *Celebration* masked
**crime** unmasked | Last year , the numbers of *celebration* applications have been adopted in **crime** since the Middle since since year has watched a rate of more than 95 per cent in the UK since 2011 , 2015 to 45 per cent \<eos\>
The Prime Minister has been a *celebration* to the course of the **crime** deal which and to have a relationship of the European Union , with the rest of the European Union has before the scandal \<eos\> |
| **War** unmasked
*play* masked | The Prime Minister David Cameron said **war** had not be hard to *play* the divorce of the European Union , and determined it would mark a divorce between the majority of the UK and leave the bloc of the European Union \<eos\>
It may be really more special **war** . . . try to *play* it , and I hope we ' re going to be able to attack this team so we have to do it \<eos\> |
| *War* masked
**play** unmasked | But I was proud of the *war* I ' ve got to **play** to it but I wish that ' s because I want to do , and I ' m pretty excited before the end of the season , he said \<eos\>
We are in the Premier Cup *war* and we want to keep play with their top six that which we need to **play** in the World Champions and at the end of season that we have to be it well \<eos\> |
| Unconstrained generations | By the time , the driver had been deployed to lie out to the incident , an new official said that the woman had not been found a them \<eos\>
According to The Wall Post survey poll found that 80 per cent of eligible older registered who , they thought he would be the rate for 10 per cent less likely to vote , while 16 per cent of those said they would still less likely to happen \<eos\> |

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
