# OpenReview forum: "SUNMASK: Mask Enhanced Control in Step Unrolled Denoising Autoencoders"
_NeurIPS.cc/2022/Conference — NeurIPS 2022 Submitted_

### Official Review · Reviewer_cQyt · 2022-07-05

**Rating:** 5
**Confidence:** 2
**Soundness:** 3 good
**Presentation:** 3 good
**Contribution:** 2 fair

**Summary:**

A generative sequence modelling based on masked unrolled denoising autoencoders, called SUNMASK, is proposed in this manuscript. The proposed separates the role of masking and correction by conditioning predictions on the mask. Compared to the baseline, the authors claim SUNMASK has a better performance on music modelling. The experiment also proves this contribution.


**Questions:**

I hope the author can clearly claim their contributions, especially to discuss the differences between SUNDAE and proposed method.

**Limitations:**

Although the experiments have shown the proposed method has great performance, the discussion about the experiment is limited. For example, Fig 6-9 is just shown on page9 without discussion.

**Strengths And Weaknesses:**

Generative modelling is a hot topic, since a lot of application can get the benefit. This idea is very close to SUNDAE proposed by DeepMind this year, which is claimed on page 6. But in Section 3.1, they mainly discuss the difference between the proposed methods and the coconut and the discussion about SUNDAE is ignored. This makes the originality not very clear. Hope the authors can give the discussion deeply and move the related\closed work to section 2. The researchers would like to know the difference when they get this manuscript since it is too closed to SUNDAE. The quality of this manuscript is good. Their designed experiments can prove their contribution and the manuscript is easy to follow. However, there are some points that could be further improved. For example, the font size of Table 1 is too small and Fig 4-9 could be shown in fig.

---

> ### Author Response · Authors · 2022-08-02
> **Reply to initial feedback from Reviewer cQyt**
>
> > Hope the authors can give the discussion deeply ..
>
> We have extended discussion on the relationship between SUNDAE, Coconet, and SUNMASK, and will continue strengthening this connection as a quick "feature list" comparison in the next draft.
>
> >Questions...
>
> A more clear discussion of related work and particularly the techniques in common between SUNMASK, SUNDAE, and Coconet has been noted by several reviewers. We have added a section specifically targeted at addressing this issue in a new draft. We have also updated the table, removing some experiments to increase font size and legibility while moving the full experiments table to the appendix, where we can enlarge the font size without space concerns.
>
> > Limitations ...
>
> We have reworked the experimental section and related discussion, to more clearly discuss the points fig 6-9 were attempting to show. Primarily, this experiment shows that the model doesn’t simply ignore the mask by treating it as a secondary noise source or random variable, and that the high trust / low trust interpretation of this input (discussed in the Method section) seems to hold in practice. Additional qualitative text experiments have been added to address this point specifically.

---

### Official Review · Reviewer_b2LK · 2022-07-10

**Rating:** 7
**Confidence:** 5
**Soundness:** 3 good
**Presentation:** 4 excellent
**Contribution:** 3 good

**Summary:**

This paper proposes a new non-autoregressive generative model for discrete sequences which combines a recent work SUNDAE with masking. The mask is provided to the network as additional conditioning - unlike e.g. BERT which masks out the tokens themselves. The advantages of the proposed technique are demonstrated on a polyphonic symbolic music generation benchmark JSB, where it outperformes another non-autoregressive method CoCoNet and the original SUNDAE. Additionally, the paper provides some experiments on text generation on EMNLP'17 News and text-8 datasets. Finally, the paper qualitatively investigates music in-painting with the proposed method. Extensive ablation studies are provided which compare different architectures and different sampling methods, e.g. finding typical sampling to be beneficial.

**Questions:**

+ Is there an AR baseline in Table 1? Is it BachMock?
+ figure 1: do correct predictions become unmasked? or do you keep mask constant for the unrolls?
+ table 1: is SMc-T better than everything else across all metrics or just overall? In the latter case, maybe use bold for the best method on the particular metric. Also, standard deviation is normally denoted as ±.
+ standard deviations in Table 1 are quite big, any way to reduce them?
+ L285-286: Negative BLEU/Self-BLEU in generated corpora of 1000 sentences - while it's understandably quite expensive to compute it for 10K samples x many temperatures (as was done in SUNDAE & ScratchGAN papers from DeepMind), keep in mind that the results are usually a bit different between 1K and 10K samples. It is certainly worth adding a disclaimer.
+ some questions (like the one about the compute) are addressed in checklist but not in the paper. Would it make sense to include those discussions in the main text?
+ Are there plans to open-source the code? Even non-runnable code is useful for people reproducing prior work.

**Limitations:**

Limitations are clear from the work itself.

**Strengths And Weaknesses:**

\+ The whole paper is well-written, illustrations are nice, music samples in the supplementary are helpful.

\+ Contribution is reasonably novel.

\+ Experiments are well-executed, with sufficient ablations to understand what changes lead to most improvement.

\+ Improvement over relevant baselines in polyphonic music generation is significant.

\- Language results are worse than SUNDAE.

---

> ### Author Response · Authors · 2022-08-02
> **Reply to initial feedback from Reviewer b2LK**
>
> > Questions...
>
> Bach Mock is an AR baseline, which also includes secondary metadata in its feature setup, primarily with respect to timing and key/pitch. This feature representation is quite different than Coconet, so to facilitate a direct comparison we use the simpler piano roll setup from Coconet. Some limitations of the simple piano roll without onsets formulation can be seen in Huang et al. [1], section 3.1, we will briefly discuss this in the experiment section as well.
>
> We think of the Bach Mock Transformer (and accompanying models from AugGen, a closely related paper to Bach Mock) as the current “upper bound” in terms of performance against this metric, since they are both designed against and utilize the metric heavily. Finding a model which uses simple piano roll, piano roll + note onset, or midi event representation while matching Bach Mock / Aug Gen in terms of metric would be a step forward in terms of research on the JSB dataset, and we see SUNMASK can be close in limited settings (Top 20, discussed more below).
>
> > figure 1...
>
> Masks are kept constant for the unrolls during training, as we wanted the model to have some “confusion” about how trustworthy the mask is - particularly if the model is poor at modeling unmasked variables (remembering the loss is masked to only focus on the masked variables at each step), it will lead to sampling errors in the first unroll step. This means that the second unroll step has ambiguity with respect to whether the input is currently correct with respect to the mask, or potentially erroneous and in need of self-correction. This provides indirect pressure for the model to correct its own mistakes, in line with the original motivations in SUNDAE.
>
> We believe this also leads to the “high trust / low trust” interpretation of the mask as discussed in the Methods section, as opposed to always trusting the mask (which would be the behavior if the mask input always matched the data after sampling). This is important, as the mask at inference is sampled randomly according to a schedule, and too much focus on treating unmasked variables as “correct” could be disastrous to overall sample quality. We have added some discussion to the draft related to this clarifying the reasoning behind this design.
>
> > table 1...
>
> Many of the high standard deviations come from bad samples which devolve into chaos, or only hold 1 chord constant (0.0 interval score but bad overall score can be an indicator of this). In the appendix we ran a secondary experiment subselecting the “top 20” out of 200 SUNMASK samples from the best overall setting, and we see extremely strong performance in terms of qualitative and quantitative results, even acknowledging the fact that selection against the metric should radically improve numerical results on that metric. Comparing to AugGen, which also did sampling and dataset selection against the Bach Mock metric, our results seem to fare a bit better. We will update the results section in the draft to include this experiment, as well as discussion about what it means.
>
> > L285-286...
>
> We will add this disclaimer, as well as continue trying to improve the performance against the strong baseline of SUNDAE / ScratchGAN. Doing a longer evaluation on the full 10k sentences is prohibitive in terms of computation but still feasible in the future, given our available hardware.
>
> > some questions (like the one about the compute) are addressed in checklist but not in the paper. Would it make sense to include those discussions in the main text?
>
> We will try to address some of the limitations in the main text, depending on space available.
>
> > Are there plans to open-source the code?
>
> We plan to make experimental code directly available, as well as some simplified code highlighting details of Coconet, SUNDAE, and SUNMASK implementations. The early forms of this were included in the supplementary material, and will be added to the github repo.
>
> [1] Music Transformer, Huang et al. https://openreview.net/pdf?id=rJe4ShAcF7

---

### Official Review · Reviewer_hRGZ · 2022-07-11

**Rating:** 3
**Confidence:** 4
**Soundness:** 2 fair
**Presentation:** 1 poor
**Contribution:** 2 fair

**Summary:**

This paper proposes SUNMASK, which models discrete sequences without ordering assumptions. The authors emphasized the fine-grained control during the iterative generation. Experiments on generating symbolic music and language model show the efficacy of the proposed SUNMASK.


**Questions:**

Figures4-9 are hard to understand, with very verbose description, maybe would be good to provide a high-level idea first?


**Ethics Review Area:**

["I don’t know"]

**Limitations:**

The authors did not really cover too much about limitations...

**Strengths And Weaknesses:**

In short, this paper is a combination of SUNDAE and Coconet. The major contribution of the paper seems to be the controlled unrolling procedure which is essentially adding a mask over the SUNDAE loss formulation. The authors include the lots of details for training and implementation which is very helpful for the readers to understand and potentially reproduce.

However, this paper has several major shortcomings:
- The organization of the writing is very loose, making it very confusing for the readers. E.g. sections are named as a mere word like “ trade-offs”, “inference”, which does not offer any additional clues to readers beside its literal meanings.
- The paper is heavily based on SUNDAE, and it is not much of an improvement compared to SUNDAE if we look at, for instance, figure 2. Since fig 2 is the same task as the original task of SUNDAE, I believe it is a apple-to-apple comparison.
- The analysis is not very convincing both quantitatively and qualitatively. Quantitatively, I think Bach Mock grading function is biased towards its own task, and it is less generaliziliby than BLEU score on the LM task, which SUNMASK does not outperform existing models. Qualitatively, the cherry-picked “ode to joy” example is good itself, but too specific, maybe it would be good we combine some human subject study?

---

> ### Author Response · Authors · 2022-08-02
> **Reply to initial feedback from Reviewer hRGZ**
>
> >The organization...
>
> For the next draft, we have focused on clarifying section titles, improving the presentation and flow of the writing.
>
> >The paper...
>
> Figure 2 is the same overall task as SUNDAE, however the resources needed to reproduce the result to perform direct model comparisons are non-trivial. Our current largest model for that task is only 200M parameters, and trains with much smaller batch sizes (20 vs 1024) given our resource limitations, which means the 8k effective steps used by SUNDAE is 400k “steps” in our model due to the batch size difference. At our current training speed (~1s per batch for transformer models in the 200M - 300M parameter range) this is too prohibitive for model development, and would require batch parallel training for meaningful progress. We will continue to work on improving the quality and speed of the text model.
>
> This is also arguably the smallest experiment in the SUNDAE paper, and the only one that seemed remotely achievable for our computational budget of ~2-3 V100 GPU, with no meaningful ability to do batch parallel training between them due to compute platform. Given recent directions in large language modeling, the sky is the limit for this kind of scaling work and work such as Bavarian and Jun et al. [1], Du and Qian et al. [2] show that the “order problem” is far from solved in language modeling. We think methods such as SUNDAE, SUNMASK, Coconet, OA-ARDM, and discrete diffusion more generally have potential to impact generation “at scale”.
>
> We also strongly outperform our best baseline SUNDAE models (which are themselves competitive with a strong previous work, Coconet) in the very small scale music setting, so that is an indication SUNMASK can improve performance on extremely resource limited tasks. Scaling and reproducing the result from SUNDAE on text data is important, and we plan to keep working toward this in the future.
>
> A secondary result of this work on SUNMASK demonstrates this overall style of modeling is not tied to architecture. Although the preceding paper methodologies imply this, all SUNDAE experiments use transformer variants, and Coconet used only convnet variations. Here we use (variably) U-Net convnets and transformers with the same SUNMASK training, as well as training SUNDAE convnets for the music experiments. This shows the loss and training formulation for these masking/denoising model variants are indeed decoupled from the underlying model type, and broadly useful.
>
>
> > The analysis...
>
> The selection of the Bach Mock metric was specifically due to the tight correlation between high Bach Mock scores and expert analysis (graduate level musicians) - the details of this are discussed in the Bach Mock paper, but we will also add a summarization in the new draft . Particularly during all model development, we did not use the metric to guide model design, rather using empirical judgements of sample quality for each task in different domains, so the end comparison between models would not be overly driven by focus on the computed metric alone, but rather the combined metric and sample quality (as judged by the paper authors throughout development). The Bach Mock metric correlates strongly with expert opinion but may have exploitable flaws if optimized against either directly or indirectly via human in the loop overfitting, similar to criticisms of BLEU in machine translation.
>
> For creative tasks specifically it is difficult to derive a broad “quality” metric that would be agreeable among subject experts, so we found the focus on only Bach-specific stylistic components to be a strength rather than a weakness. Broad user studies on the quality of infilling / harmonization can be useful (see Huang et al. [3]), but are heavily affected by which users are targeted and their background or desires in such a tool. As such, we decided to avoid human studies and larger interaction research for this particular submission, choosing the Bach Mock metric to see whether structural components of Bach’s music were successfully learned by the generative model.
>
> >Questions...
>
> We have reduced this figure, and added a text equivalent to this experiment as well as a dedicated section discussing this particular experiment and demonstration of the usefulness of the mask variable.
>
> >Limitations...
>
> We will move some of the limitation discussions from the appendix, into the main document, given space is available.
>
> [1] Efficient Training of Language Models to Fill in the Middle, Bavarian and Jun et al. https://arxiv.org/abs/2207.14255
>
> [2] GLM: General Language Model Pretraining
> with Autoregressive Blank Infilling, Du and Qian et al. https://arxiv.org/abs/2103.10360
>
> [3] Bach Doodle: Approachable music composition with machine learning at scale, Huang et al. https://arxiv.org/abs/1907.06637

---

> > ### Comment · Reviewer_hRGZ · 2022-08-09
> > **Thank you for your replies!**
> >
> > Dear Authors:
> > I appreciate your replies and I have read the reviews of other reviewers as well.
> > Wish you good luck revising the current draft! I believe the paper will be of value to the community in its full form.
> > As the current rebuttal window is closing, since I haven't seen a revision or an opensourced code base. I remain my initial score.

---

### Official Review · Reviewer_EWiy · 2022-07-12

**Rating:** 6
**Confidence:** 1
**Soundness:** 3 good
**Presentation:** 3 good
**Contribution:** 3 good

**Summary:**

This paper introduced a new generative modelling method for structured data and demonstrated its effectiveness on symbolic polyphonic music and character level and word-level text modelling. This method also offers the opportunity for subtle control at inference time, which could be useful for downstream human-in-the-loop applications.

**Questions:**

In Table 1, sometimes SD-AT seems to be doing better (e.g. for Note and Rhythm and slightly better for Repeated Sequence”, why is SD-AT not bolded for those and instead SMc-T is bolded? I understand that overall SMc-T performed the best, but it’s a bit confusing to me because bolded numbers are not always the best.

DO you think including Figure 3 in the main paper would be necessary? Personally it is very hard for me to imagine the sound merely by looking at the visuals. A project page with generated music would be a much better media to demonstrate this, but given the limitation of the paper format, perhaps those figures could be moved to the appendix. With the space saved, authors could consider including more details about closely related works such as SUNDAE and Coconet.


**Limitations:**

Yes, authors have discussed in detail the ethical implications of their work in appendix section A.

**Strengths And Weaknesses:**

Strengths

Originality: This paper seems fairly original to me, but I am not sure since I am not familiar with prior work.

Significance: This paper seems fairly significant to me as author’s method is overall the best compared to other closely related works (even though I don’t know whether those are SOTA methods on this task).

Quality: This work seems to be of high quality as there seems to be significant engineering effort put into the models to make them work well. I always have respect for such engineering efforts.

Weaknesses

Clarity: To me, some places in this paper feel a lack of context. Since this paper is built on prior works like SUNDAE and Coconet, it’d be better to detail those two important prior works. This could be used to replace sections 1.1 and 1.2, as most people in NeurIPS community should at least know the gist of autoregressive models and non-autoregressive models. However without the context of SUNDAE and Coconet, it is hard to understand how this method differ from those works and the new insights in this work.

---

> ### Author Response · Authors · 2022-08-02
> **Reply to initial feedback from Reviewer EWiy**
>
> >Clarity...
>
> A more clear section, summarizing the key points of SUNDAE and Coconet, as well as the relation to SUNMASK and particularly which pieces of each method are used, has been added to the paper. We have also tightened the discussion on AR and NAR models, freeing more space for other writing.
>
> >In Table 1...
>
> We have updated the bolding of the chart to highlight the best on each sub metric, and will add discussion around this result and the underlying metric. Bolding only the best overall performance oversimplified the result, but the Bach Mock metric was also created as a compound metric, with various weightings across each of the sub metrics (such as note, rhythm, etc) in order for the final value to correlate strongly with expert analysis. Particularly, the fact that the overall best model in terms of median performance does not need to be the best on any particular sub metric is a direct feature of the overall metric design of Bach Mock.
>
> >Do...
>
> We have reduced the figures from Figure 4-9 and combined with figure 3, as well as the size of figure 3. and used the space to discuss in more detail added experimental results. The project website was not complete during the initial submission phase, but we have now updated the git repository and web presentation to include a midi sample player for the samples which were previously contained in the supplementary materials. Having a few visual examples of the piano roll is useful as many other works on piano roll generation show these visually as well, however they were too domineering in the first submission draft and have been reduced in the new draft.

---

> > ### Comment · Reviewer_EWiy · 2022-08-10
> > **Response to the rebuttal.**
> >
> > I have read the author’s response, the updated paper and other reviewers’ responses and have decided to keep my score. In hindsight, after reading other reviewers’ reviews, I think I should’ve given a five, it seems most reviewers agreed that the writing is not good enough for publication and the significance is not enough given prior works. However, the authors have made a lot of improvements after the initial submission and have addressed all my concerns, and that made me decide to “raise” my five to a six, so in aggregate the decrease and the increase cancel out and I think six is my most accurate estimate of the quality of this submission.

---

### Official Review · Reviewer_y5UN · 2022-07-13

**Rating:** 3
**Confidence:** 2
**Soundness:** 2 fair
**Presentation:** 1 poor
**Contribution:** 1 poor

**Summary:**

Authors propose a diffusion model that adds a mask as an additional inputs to noisy data. The mask controls which positions in the input should be denoised, in order to provide control for partial inference with controllable positions. Authors experiment on polyphonic music modeling and language modeling.

**Questions:**

* Please number equations
* The github repository linked in appendix is empty
* L110: why introducing m_t as a time-dependent mask while as soon as L116 it’s limited to a constant mask?
* L160: is there a specific motivation for sampling the difficulty variable at the batch level?
* L169: why not using the same notation as in the loss equation?
* L172: schemes -> scheme
* L185: or -> for
* L186: trained model -> the trained model
* L213: papers to use -> papers use
* L306: underflying -> underlying

**Limitations:**

Authors adress ethical concerns of music and language modeling in Appendix A, and current inference runtime issues in Appendix F.

**Strengths And Weaknesses:**

# Strengths:
* The task of controllable generation is interesting, particularly to allow humans to interact with generative models. Allowing users to choose which pieces to improve is a particularly simple yet effective way of providing intuitive control and interaction.


# Weaknesses:
* The writing could be considerably improved. In particular, the introduction is very long and has several flaws such as:
  * The first paragraph is too vague to be of any use and I recommend authors remove it
  * The role of the introduction is to give context around the paper’s contribution, why it is useful and why the author's method is worth considering. However, the motivation of authors and the logical premises to their contribution are completely absent of the introduction. The introduction should at least finish with something akin to the first paragraph of the method section, which introduces the contribution in context.
* The method section is written such that a reader who is not an expert with SUNDAE (a previous work which we cannot expect the average ML researcher to know) will need to go back and forth between papers. While I acknowledge there are several opinions on this, I personally think the paper should be as self-contained as possible, and should provide much more context around the method.
* The motivation of the authors is not clear to me: unrolling and using a time-dependent mask are presented as key components of the method, yet the model is only used with 2 steps (without ablation) and a constant mask, which is a bit underwhelming and does not support author’s claims that unrolling is useful (claim in L221). Ablation studies would be a minimum.
* Experiments are performed on three datasets, yet their presentation is minimal. Results on polyphonic music modeling are just presented as a big table with zero comment on the metric, the baselines, and the variants of the model. Similarly, experiments on the news dataset are not commented on, baselines shown in the plot are not even cited. Authors should definitely revamp the experiments section to better highlight the advantages of their method.

In summary, while the topic is of wide interest for NeurIPS, the presentation of the author’s method and experiments is currently too flawed to be worth publication at NeurIPS.

---

> ### Author Response · Authors · 2022-08-02
> **Reply to initial feedback from Reviewer y5UN**
>
> >The writing...
>
> Thank you for the writing feedback, we have updated the introduction section, tightening the overall presentation, with a focus on linking the proposed method more to the related background.
>
> >The method...
>
> Given this suggestion alongside the other review comments, we have added a dedicated section comparing SUNDAE, Coconet, and SUNMASK in more detail, focused on the commonalities and differences across all 3 methods.
>
> >The motivation...
>
> 2 step unrolling was found to give the best performance in SUNDAE, and given this extensive ablation study and our practical limitations we set this hyperparameter in the same manner after early experiments. Notably, each increase in the number of unrolling steps is effectively a multiplier on the batch size, and unrollings of more than 3 or 4 become prohibitive in terms of memory usage during training without large scale batch parallel training or other changes to improve memory efficiency. Comparisons against Coconet directly show the benefits of unrolling as well, as Coconet does not have an unrolled loss. We will explicitly mention how this hyperparameter is set in the text.
>
> The constant mask is used during training but during inference the mask is continually changed over timesteps. Alternatives during training are possible, but holding the mask constant over both steps creates ambiguity the model must resolve (as discussed in the Methods section). Changing masks at each step during training has the potential of creating excess randomness (in which case the model may learn to ignore the mask), or too exact information (in which case the model will trust the mask completely, resulting in worse inference samples). Extensions using model probabilities, confidences, or secondary domain information for training and inference masks are a potential direction for follow-up work. We have added a demonstrative experiment on language in order to show the effect of the mask more directly, with a focus on the “high trust/low trust” interpretation.
>
> >Experiments...
>
> Discussion on the baselines and variants of music models from the figure caption has been moved to a dedicated space of the paper, as well as some information about the motivation behind each of the experiments (for both music and text). We generally focus on moving information out of the figure captions and into the text body in the updated draft. The absence of the baselines from our citations is an egregious oversight, we have added all baselines compared against in the revised draft and thank the reviewer for catching that mistake.
>
> >L110:...
>
> We currently write the equation in full generality, allowing different masks at different steps, clarifying that in these experiments m_t is held fixed during training. This presentation also creates consistency with the inference procedure, where the input mask m_t (which is also the loss mask) is varied at every step.
>
> >L160: ...
>
> Difficulty level being sampled at the sequence level (and specifically a step level random variable compared against sequence level) derives from the SUNDAE training scheme, particularly the relation to discrete diffusion. There is no direct “batch level” noise (only sequence / step level), as we want different noise levels per sequence to be mixed together in every batch for better training stability. This is discussed in the body and appendix of SUNDAE, appendix A respectively, derived from the work of Hoogeboom et al. [1] as well as the paragraph following l160. We have added a small portion of this motivation to the model comparison section in order to help the SUNMASK manuscript be more standalone as noted in the feedback.
>
> The particular sentence in question (160) is misleading, as none of the noise operates on a minibatch level (beyond the mechanics of feeding into the model) and we will update the wording to better reflect the noise sampling procedure.
>
> >L169: why not using the same notation as in the loss equation?
> ...
>
> These mistakes and inconsistencies have been corrected in the updated draft.
>
> [1] Hoogeboom et al. Argmax Flows and Multinomial Diffusion: Learning Categorical Distributions
>  https://arxiv.org/abs/2102.05379

---

### Author Response · Authors · 2022-08-02
**Thank you to all reviewers**

We would like to thank all reviewers for taking the time to analyze and suggest improvements to our manuscript.

Reviewers generally agreed on these primary areas to target: tightening of the introduction and overall writing presentation, better direct comparison between SUNDAE, Coconet, and SUNMASK in terms of methodology, and more exposition on the experiments. We have uploaded a revised draft of the paper to better address these points, as well as adding additional qualitative text experiments to better analyze the role of the mask in generation.

In the interest of space, we have removed the text8 experiment, which was not previously compared to other character level diffusion models, and plan to add that comparison in the appendix. The website and github [https://github.com/SUNMASK-web/SUNMASK](https://github.com/SUNMASK-web/SUNMASK) will be updated and improved continually throughout the reviewer/author feedback process. We will also continue to update the paper draft, in order to better fit the information desired based on reviewer feedback into the page limit.

Individual replies to each reviewer, discussing separate concerns will added as directed comments.

---

### Meta-Review · Area_Chair_t5z2 · 2022-08-20

**Recommendation:** Reject
**Confidence:** Certain

**Metareview:**

This paper introduces SUNMASK for modeling discrete sequences. It builds upon previous works such as SUNDAE, Coconet and order-agnostic NADE, but uses a masking scheme that enables fine-grained or human-in-the-loop control during the generation. The qualitative experiments about musical inpainting and masking terms in language modeling do support this motivation to some extent. However, the reviewers are mainly concerned with both the algorithmic novelty and experiments of the paper.

Regarding the algorithmic novelty, some reviewers are concerned that the method is a straightforward combination of SUNMASK and Coconet. I tend to agree with this. Also the reviewers are concerned that the paper could have done a better job in the introduction and background section by putting the method in a better context and better describing related methods such as SUNDAE. This could help highlight the novelty of the paper.

Regarding the experiments, some reviewers are concerned about language modeling (Fig. 2) experiments, and that they do not show much improvement over SUNDAE. I tend to agree, and this is also shown in the results of Table 3 where it is clear that the quality of the language model is not on a par with the recent developments.

Also some of the motivations of the paper, especially the arguments about "high trust / low trust" interpretation of the mask, was unclear to me.

In short, I believe the paper should be clarified and improved by addressing the above concerns.

**Award:**

No

---

### Decision · Program_Chairs · 2022-09-14

Reject